# Strong relation between atmospheric CO<sub>2</sub> growth rate and terrestrial water storage in tropical forests on interannual timescales

Samantha Petch<sup>1</sup>, Liang Feng<sup>2</sup>, Paul Palmer<sup>2</sup>, Robert P. King<sup>3</sup>, Tristan Quaife<sup>1</sup>, and Keith Haines<sup>1</sup>

**Correspondence:** Samantha Petch (s.petch@pgr.reading.ac.uk)

**Abstract.** The atmospheric CO<sub>2</sub> growth rate (CGR) is characterised by large interannual variability, mainly due to variations in the land carbon uptake, the most uncertain component in the global carbon budget. We explore the relationships between CGR and global terrestrial water storage (TWS) from the GRACE satellites. A strong negative correlation (r = -0.70, p <0.01, based on monthly data) between these quantities over 2002-2023 indicates that drier years correspond to a higher CGR, suggesting reduced land uptake. We then show regional TWS-CGR correlations and use a metric to assess their contributions to the global correlation. The tropics can account for the entire global TWS-CGR correlation, with small cancelling contributions from the Northern and Southern Hemisphere extratropics. Tropical America makes the dominant contribution (69%) to the global TWS-CGR correlation, despite occupying < 12% of the land surface. Aggregating TWS by MODIS land cover type, tropical forests exhibit the strongest CGR correlations and contribute most to the global TWS-CGR correlation (39%), despite semi-arid and cropland/grassland regions both having more interannual TWS variability. Tropical forests exhibit the strongest CGR correlations due to their high productivity and sensitivity to water stress, which strongly influences interannual variations in carbon uptake. An ensemble mean of eight atmospheric CO<sub>2</sub> flux inversion products also indicate a 66% tropical contribution to CGR variability, with tropical America/Africa contributing 27%/28% respectively. Regarding land cover type, semi-arid/tropical forests contribute almost equally (37%/35%) to CGR variability, although tropical forests cover a much smaller surface area (25%/10%). Time series of global and regional TWS and CO<sub>2</sub> flux inversions through 2002-2023 also show changing regional contributions between global CGR events, which are discussed in relation to regional drought and ENSO events. Our study advances previous work by providing a more detailed analysis of regional contributions and doing a temporal breakdown of contributions.

#### 1 Introduction

The atmospheric CO<sub>2</sub> growth rate (CGR) exhibits large amounts of year-to-year variability, which holds high significance in the context of climate change mitigation and projection. This variability is predominantly driven by fluctuations in the land carbon uptake, which has a year-to-year variability of about 1 GtCyr<sup>-1</sup>, with smaller contributions from oceanic uptake and anthropogenic emissions (Friedlingstein et al., 2023). The land carbon sink results from an imbalance between the uptake of carbon through photosynthesis (GPP), the loss of carbon through ecosystem respiration (ER), and carbon losses through

<sup>&</sup>lt;sup>1</sup>National Centre for Earth Observation, Department of Meteorology, University of Reading, Reading, RG6 6BB, UK

<sup>&</sup>lt;sup>2</sup>National Centre for Earth Observation, School of GeoSciences, University of Edinburgh, Edinburgh, EH9 3FF, UK

<sup>&</sup>lt;sup>3</sup>Hadley Centre, Met Office, Exeter, EX1 3PB, UK

other disturbances (D), such as fire and land use change. The net carbon flux is termed the net biome production (NBP), where NBP = GPP - ER - D. This land carbon sink plays a crucial role in offsetting anthropogenic CO<sub>2</sub> emissions, accounting for approximately 30% of these emissions each year (Friedlingstein et al., 2020). However, NBP remains the most uncertain component of the global carbon budget, with modelling studies indicating significant uncertainties in both the magnitude and sign of future projections of the sink (Ahlström et al., 2012). Reducing these uncertainties will require a better understanding of the processes underlying the CO<sub>2</sub> fluxes and how they might change in the future, which will be essential for shaping effective mitigation policies.

It is well documented that the CGR interannual variability (IAV) is closely related to El Niño-Southern Oscillations (ENSO) (Gurney et al., 2012). There are noticeable increases in CGR during El Niño events and decreases during La Niña events (Keeling and Revelle, 1985). There is also a widespread consensus that variations in tropical ecosystems exert the most significant influence on global CGR, with numerous studies concentrating their efforts on the Tropics (e.g., Wang et al., 2013, 2014; Liu et al., 2023; Luo and Keenan, 2022). In particular, tropical temperature is found to have a strong positive correlation with CGR (Wang et al., 2013). Other physical factors governing the land carbon uptake remain elusive. Given that tropical temperatures are typically considered optimal for photosynthesis (Huang et al., 2019), any elevation in these temperatures are anticipated to have a dampening effect on GPP while simultaneously elevating ER. This combined impact may amplify the role of temperature as a significant driver of CGR variability.

While many studies have pointed to the influence of temperature on CGR (e.g., Cox et al., 2013; Wang et al., 2013), recent research is increasingly highlighting the significance of water availability as a primary control. The launch of the Gravity Recovery and Climate Experiment (GRACE) satellites in 2002 has largely helped to support this notion by providing accurate terrestrial water storage (TWS) data based on gravitational measurements. Humphrey et al. (2018) used the GRACE data and demonstrated that there is a strong negative correlation between CGR variability and observed changes in global TWS (monthly r = -0.65 and yearly r = -0.85), revealing that drier years, especially in the tropics, coincide with higher CGR. The study also highlighted that this water storage relationship is clearer than for precipitation, as previous studies find there is a weaker and lagged response to precipitation. Most vegetation responds to soil moisture (Wang et al., 2016), which is a component of TWS, whereas precipitation anomalies only account for water input and do not consider losses from evapotranspiration or runoff. Even when combined, water flux observations inadequately capture the interannual variations observed by GRACE (Petch et al., 2023a). He et al. (2022) demonstrated that interannual variability in atmospheric VPD was also significantly negatively correlated with net ecosystem productivity and hence to the interannual variability of the CGR, although this analysis relied upon VPD estimates from FluxCOM and TRENDY. In this study, we focus on the influence of TWS on CGR due to the availability of large-scale, observation-based data from the GRACE satellite mission. In contrast, VPD at global or continental scales is typically derived from model-based products or reanalysis data. While TWS is not necessarily a better explanatory variable to temperature or VPD, it offers a complementary perspective and allows us to leverage independent observational datasets.

Separating the individual contributions of various climatic drivers to CGR fluctuations can be challenging due to their often intertwined nature. For example, VPD and soil moisture are coupled, where high VPD usually corresponds to dry soils and

low VPD to wet soil. As a result, vegetation responses to water stress are likely to reflect the combined influence of both variables, making their individual effects difficult to disentangle. Temperature and water availability are also interrelated, further complicating attribution. Nevertheless, Humphrey et al. (2018) showed that the relationship between CGR and TWS stands independently of temperature influences, suggesting that TWS captures a distinct and meaningful component of CGR variability.

65

Wang et al. (2022) then showed that the relative influence of temperature and water on net ecosystem exchange (NEE) IAV shifts across different regions and seasons. They employed three different approaches: atmospheric inversions, process-based vegetation models from TRENDY, and a data-driven model (FLUXCOM). Their findings reveal broad agreement that the tropics are a key driver of global correlations; however, the dominant driver of global NEE IAV varied due to disagreements regarding the seasonal temperature effects in the Northern Hemisphere. This underscores the critical importance of understanding the relative magnitudes of water and temperature contributions in the Northern Hemisphere for determining the dominant drivers of NEE IAV. Another recent study by Liu et al. (2023) provides evidence that the coupling between interannual CGR and TWS is becoming increasingly strong. They report an increase of around 35% in CGR sensitivity to tropical water variations from 1989-2018 compared to 1960-1989, however, existing climate models do not exhibit signs of this rising trend in tropical water-CGR coupling (Liu et al., 2023). Models are also found to have a tendency to underestimate the strength of the coupling (Humphrey et al., 2018). In general, interactions between TWS variations and the carbon cycle are a key uncertainty in current climate models that could strongly influence CGR over the coming decades.

Despite the dominant role of terrestrial ecosystems in influencing CGR variability, the heterogeneity of these ecosystems means pinpointing the specific land regions contributing to IAV can be challenging and there are inconsistencies among previous studies concerning the land cover types contributing to IAV. Ahlström et al. (2015) used process based models and found that semi-arid ecosystems could explain 39% of the global NBP variability over 1982-2011, thereby exerting dominant control over carbon sink interannual fluctuations, while the mean sink was primarily influenced by tropical forests. Zhang et al. (2016) found that semi-arid regions contributed 57% of the detrended interannual variability in global GPP. These results are supported by Humphrey et al. (2018), who used GRACE to demonstrate that TWS in semi-arid regions contribute the most to the global storage variability, while also mentioning a possible role of TWS in tropical forests which is also well correlated with the CGR.

Marcolla et al. (2017) examined the IAV in the terrestrial carbon budget using three different datastreams: terrestrial ecosystem level observations from FLUXNET (La Thuile and 2015 releases; Baldocchi et al., 2001), a bottom-up global product resulting from upscaling site level fluxes (the MPI-MTE; le Maire et al., 2010), and a top-down inversion product (Jena Carbonscope; Rödenbeck et al., 2003). They found significant discrepancies among these datastreams regarding the main sources of temporal variability, particularly in the tropics, where there is a lack of atmospheric and ecosystem observations. However, all products unanimously identified several crucial global features, in particular the high relative IAV in the terrestrial carbon cycle in water-limited ecosystems. Similarly, Piao et al. (2020) used three different approaches, including land carbon cycle models, a data-driven model (FLUXCOM; Jung et al., 2019), and atmospheric inversion models (taking the mean of CAMS and Jena CarbonScope; Chevallier et al., 2010; Rödenbeck et al., 2003), to investigate regional contributions to global terres-

trial carbon IAV. They found the share in contributions of tropical semi-arid regions versus tropical non-semi-arid regions was similar between approaches. However, they found relatively larger contributions from the extratropics in atmospheric inversions compared to other approaches, possibly due to limited surface CO<sub>2</sub> observations over the tropics. These data are used exclusively in the atmospheric inversions, affecting their ability to discern IAV between the tropics and the extratropics.

Our research first aims to assess the global relationship between TWS and CGR interannual variability, extending Humphrey et al. (2018) analysis in time from 2016 up until 2023 and highlighting the global CGR variability we are seeking to explain. This period notably includes some of the largest values of CGR on record. We then look to regionalise the contributions based on both land cover type and large spatial regions, and look at interannual variability events through the whole period. Additionally, we use eight atmospheric CO<sub>2</sub> inversion products to infer regional contributions of CO<sub>2</sub> fluxes. This approach allows us to make similar assessments without relying on assumptions of the relationships between water and CGR. We assess the agreement between the inversion products and GRACE, as well as the consistency among inversion products. We also look at major temporal events in the CGR record to examine which regions are contributing most at different times and assess the consistency between these approaches. We also use the inversions to estimate regional-scale sensitivity of NBP to TWS variability.

Section 2 discusses the data and analysis methods we use to attribute the regional contributions to global signals. In section 3, we examine the relationships between GRACE TWS on global and regional scales, and the CGR signal on interannual timescales from 2002-2023. Section 4 looks at eight atmospheric CO<sub>2</sub> inversion models from GBC2023 (Friedlingstein et al., 2023) and their ability to regionalise the terrestrial sources of CO<sub>2</sub> over the same time period. Section 5 discusses these results in relation to meteorological conditions through the period, and section 6 provides a summary and conclusions.

## 2 Data and Methods

## 115 **2.1 GRACE**

100

105

Terrestrial water storage data are obtained from the Gravity Recovery and Climate Experiment (GRACE) and its successor mission, GRACE Follow-On (GRACE-FO). The version used here is the Jet Propulsion Laboratory (JPL) Mascon RL06v2 (Wiese et al., 2016). This dataset contains gridded monthly global water storage/height anomalies relative to a time-mean, derived from GRACE and GRACE-FO and processed at JPL using the Mascon approach. The water storage/height anomalies are given in equivalent water thickness units (cm). This version of the data employs a Coastal Resolution Improvement filter that reduces signal leakage errors across coastlines. Data can be obtained from http://grace.jpl.nasa.gov/data/get-data/jpl\_global\_mascons/. There were a small number of months with missing data which were filled with monthly climatology plus the temporal interpolation of monthly storage anomalies.

## 2.2 GML surface flasks

Measurements of atmospheric CO<sub>2</sub> concentration are taken from the Global Monitoring Laboratory (GML) of the National Ocean and Atmospheric Administration (NOAA) (Lan et al., 2023). This dataset compiles measurements from the Cooperative Global Air Sampling Network, where air samples are collected approximately weekly from a globally distributed network of sites. Data can be found at https://gml.noaa.gov/ccgg/trends/global.html.

## 2.3 Atmospheric CO<sub>2</sub> inversions

This study uses eight different atmospheric CO<sub>2</sub> inversion products, summarised in Table 1, selected based on their temporal coverage, all of which span at least from 2002 to 2022. These products are from GCB2023 where the land biosphere fluxes have been adjusted to a common fossil fuel emissions dataset (Friedlingstein et al., 2023). Other similar products were discounted on the grounds they only cover shorter periods. Top-down inverse models provide spatially and temporally resolved estimates of the net CO<sub>2</sub> flux exchanged between the surface and the atmosphere. They are generated as source/sink solutions using atmospheric transport models made to fit surface flask based atmospheric CO<sub>2</sub> mole fraction measurements within their uncertainties, at various locations (Ciais et al., 2022). Typically, these inversion products employ a Bayesian statistical approach, where an optimal surface CO<sub>2</sub> flux is determined as the maximum likelihood estimate within a statistical distribution of possible fluxes. The products also use prior surface flux values and their associated uncertainty distributions. We also examined the interannual variability in the prior fluxes, where available, and in all cases found very little relationship with the posterior fluxes, and hence conclude that the prior choices were not directly influencing our results.

Atmospheric CO<sub>2</sub> inversion products differ in their use of transport models, meteorological inputs, and prior flux assumptions—all of which can significantly influence flux estimates, particularly when comparing tropical and extratropical regions (Peylin et al., 2013; Chevallier et al., 2010). In the tropics, flux estimates are especially sensitive to how transport models represent deep convection and vertical mixing, while extratropical estimates are more influenced by synoptic-scale advection and boundary layer dynamics. Prior flux assumptions, such as prescribed seasonal cycles or vegetation responses, can also introduce regional biases—especially in the tropics where these assumptions often fail to capture complex climate—ecosystem interactions (Munassar et al., 2022; Gaubert et al., 2019). Moreover, the relative scarcity of CO<sub>2</sub> observations in tropical regions amplifies the impact of model structure and prior uncertainty, whereas denser observational networks in the extratropics provide stronger constraints on inversion results (Patra et al., 2005; Schuh et al., 2019).

The UoE and CT NOAA products both adopt an ensemble Kalman filter (EnKF) approach, whereas NISMON-CO<sub>2</sub> employ a BFGS-based quasi-Newton method. CAMS on the other hand uses a variational approach and Jena CarboScope employs a Bayesian inversion framework. For the prior terrestrial fluxes UoE and CT NOAA use the CASA biogeochemical model introduced by Potter et al. (1993), NISMON-CO<sub>2</sub> use data from the Vegetation Integrative SImulator for Trace gases (VISIT) (Ito, 2019), whereas the land priors are climatological in the CAMS product. MIROC uses both CASA and VISIT Chandra et al. (2021), and Jena Carboscope uses the LPJ biosphere model (Sitch et al., 2000). The 8 products share harmonized fossil fuel and fire emissions, all using the same version of the Global Fire Emissions Database (GFED) (van der Werf et al., 2017).

**Table 1.** Summary of atmospheric CO<sub>2</sub> flux inversion products.

| Product                | Transport model | Meteorol. fields | Fossils         | Reference            |
|------------------------|-----------------|------------------|-----------------|----------------------|
| CAMS                   | LMDZ            | ERA5-Interim     | GCP-GridFED     | Chevallier (2023),   |
|                        |                 |                  |                 | Chevallier et al.    |
|                        |                 |                  |                 | (2005)               |
| UoE                    | GEOS-Chem       | MERRA2           | Oda and Maksyu- | Feng et al. (2009),  |
|                        |                 |                  | tov (2011)      | Feng et al. (2016)   |
| NISMON-CO <sub>2</sub> | 1 NICAM-TM      | JCDAS, JRA-55    | GCP-GridFED     | (Niwa et al., 2022), |
|                        |                 |                  |                 | Niwa et al. (2017)   |
| NOAA Carbon-           | TM5             | ERA5-Interim     | Millier, ODIAC  | Jacobson et al.      |
| Tracker                |                 |                  |                 | (2023)               |
| CarbonTracker Eu-      | TM5             | ERA-Interim      | EDGAR4.2        | van der Laan-        |
| rope (CTE)             |                 |                  |                 | Luijkx et al. (2017) |
| Jena CarboScope        | TM3             | NCEP reanalysis  | EDGAR           | Rödenbeck et al.     |
|                        |                 |                  |                 | (2003, 2018)         |
| IAPCAS                 | GEOS-Chem       | GEOS-5           | Oda and Maksyu- | Feng et al. (2016);  |
|                        |                 |                  | tov (2011)      | Yang et al. (2021)   |
| MIROC4-ACTM            | MIROC4-ACTM     | JRA-55           | GridFED         | Chandra et al.       |
|                        |                 |                  |                 | (2021)               |

## 2.4 MODIS

Land cover types are classified using the Terra and Aqua combined Moderate Resolution Imaging Spectro-radiometer (MODIS) Land Cover Climate Modelling Grid product MCD12Q1 Version 6.1, available from https://lpdaac.usgs.gov. We use the International Geosphere-Biosphere Programme (IGBP) legacy classification schemes, and group into six surface classes, shown in Supplementary Figure 1. These will be used to determine aggregated flux information.

## 2.5 Data processing and analysis framework

The CGR is derived by taking the differential of monthly atmospheric CO<sub>2</sub> concentration data. We calculate the interannual variability by first removing the mean seasonal cycle from the monthly data, then removing the linear trend. We assume that by removing the linear trend this removes the main fossil fuel driving signal. However, this could also remove any trend in biogenic and ocean fluxes. The CGR time series and CO<sub>2</sub> inversion products are then smoothed with a 12 month moving mean, while the GRACE TWS data is smoothed using a 6 month moving mean, following (Humphrey et al., 2018). Different smoothing windows are used because the CGR data is inherently noisier than the TWS data, therefore requiring slightly stronger smoothing to reveal meaningful interannual variability. To convert the global CGR from ppm to GtC we use the conversion

factor 1 ppm by volume of atmosphere  $CO_2 = 2.13$  GtC. We generate an ensemble mean of inversion products spanning from January 2002 to December 2022, with each product equally weighted.

To assess the importance of different regions to a global total, we use the "contribution index"  $(f_j)$ , as defined by Ahlström et al. (2015) also adopted by Zhang et al. (2019) and Humphrey et al. (2018). This index quantifies the spatial contributions to the global monthly time series t, scoring regions based on how consistent their regional flux IAV is with the sign and magnitude of the global IAV. It is given by:

$$f_j = \frac{\sum_t \frac{x_{jt}|X_t|}{X_t}}{\sum_t |X_t|} \tag{1}$$

where  $x_{jt}$  represents the detrended regional data j at time t, and  $X_t = \sum x_{jt}$  denotes the detrended global value. Regions with higher positive values of  $f_j$  contribute more to the global variations. This is used on both the regional GRACE TWS variations, and on regional CO<sub>2</sub> fluxes from the atmospheric inversion products. Regions may represent continental areas or areas with particular land cover classes depicted in Supplementary Figure 1.

To connect the TWS and CGR time series we use the Pearson correlation coefficient (r). Additionally, we look to identify the regional origin of the global TWS (GTWS) correlation with CGR. Let GTWS represent the global TWS signal and TWS<sub>i</sub> represent the storage signal at grid point i. The relationship between the global correlation  $r_{GTWS,CGR}$ , and grid point correlations  $r_{TWS_i,CGR}$ , can be expressed as;

185 
$$r_{GTWS,CGR} \times \sigma_{GTWS} \times A = \sum_{i} r_{TWS_i,CGR} \times \sigma_{TWS_i} \times a_i,$$
 (2)

where  $a_i$  is the area of grid box i, A is the total land surface area such that  $A = \sum_i a_i$ , and  $\sigma_{GTWS}$  and  $\sigma_{TWS_i}$  are the temporal standard deviations of the global and grid point storage signals, respectively. This equation holds due to the additivie properties of covariance. Hence, the contribution of a particular region (R) to the global GTWS-CGR correlation, which we will denote as q, can be expressed as a percentage as follows;

$$190 g(R) = \frac{\sum_{i \in R} a_i \times \sigma_{TWS_i} \times r_{TWS_i,CGR}}{A \times \sigma_{GTWS} \times r_{GTWS,CGR}} \times 100. (3)$$

This helps identify regions which may have a smaller contribution to the GTWS variability but which are more strongly correlated to CGR variations. This is similar to the metric used by Wang et al. (2022) in their equation (4).

# 3 CO<sub>2</sub> Growth Rate (CGR) and Terrestrial Water Storage (TWS)

175

Extending the results from Figure 1a of Humphrey et al. (2018), our Figure 1 shows the de-trended time series of global terrestrial water storage from GRACE (note the reversed axis direction for TWS), and the CGR derived from surface flasks, for the period January 2002 to December 2023. It is important to note that, although atmospheric CO<sub>2</sub> concentrations continue to

increase each year, the CGR has been detrended; therefore, negative CGR anomalies reflect periods of below-average growth relative to the long-term mean growth rate. A negative correlation of r = -0.70 (p 

**Figure 1.** Comparison of the interannual variability of global terrestrial water storage (GTWS) from GRACE (right inverted axes) and atmospheric CO<sub>2</sub> growth rate (CGR) (left axes) from 2002 to 2023. Time series have been de-seasoned and de-trended. The grey dashed line marks the end of the Humphrey et al. (2018) analysis.

To highlight smaller scale features of this relationship, Figure 2a shows the monthly correlation between GRACE TWS IAV at each grid point and the global CGR time series. The map has been filtered to only show correlations significant at the level of p < 0.01, which corresponds to a minimum correlation coefficient of  $r = \pm 0.16$ . We also calculated the map using the Spearman's correlation coefficient (results not shown) and the spatial patterns were the same as in Figure 2a. The most prominent negative correlations appear over the eastern Amazon and extend across eastern parts of tropical South America. Other areas of negative correlation in the tropics extend through India, Southeast Asia and northern Australia. There is also a sub-

stantial region in northeastern Siberia showing strong negative correlation. However, not all areas with high local correlations contribute meaningfully to the total GTWS–CGR relationship. This is often due to either a small magnitude of TWS variability or a limited spatial extent, which reduces their overall influence on the global signal. To address this, we evaluate the contribution of each 1°×1° grid cell to the global GTWS signal using Equation 1, as shown in Figure 2b. This contribution metric effectively downweights regions with minimal influence due to low TWS variance allowing a clearer assessment of the regions that dominate GTWS IAV. In this representation, tropical areas stand out much more strongly, as the metric incorporates the magnitude of TWS variability in addition to correlation strength.

From Figure 2, it can be seen that in certain regions, TWS variability exhibits positive correlations with the global CGR, such as in South Brazil and East China. There could be possible biophysical explanations for this result, for instance, it was suggested that the Brazilian Southeast is a transition region characterized by rainfall anomalies with opposite signals related to ENSO (Coelho et al., 2002). It is also possible that these regions simply do not contribute substantially to the global CGR, and the observed positive correlations may reflect local or regional processes that are largely cancelled out at the global scale. Determining the underlying causes of these correlations would require reliable estimates of regional-scale carbon fluxes, but as will be discussed in Section 4, atmospheric CO<sub>2</sub> inversion products currently do not do a good job at capturing fine scale interannual variability, limiting their utility for such attribution.

Figure 2. (a) Correlation map of GRACE TWS and CGR for the period 2002-2023. Only areas with p 

**Figure 3.** (a) Correlation between TWS signal for different regions and global CGR is represented by the blue bar on left axis. Contribution of continental/hemispheric regions to global TWS-CGR correlation is shown in orange, and contribution to global TWS signal is shown in red against the right axis. (b) This shows the contributions from each land cover type, using the same colour key. The dashed line represents the global TWS-CGR correlation.

surface area, largely contribute to GTWS signal (f = 22%). However, they have a relatively low correlation with CGR and overall do not contribute much to the GTWS-CGR correlation (g = 15%). Tropical forests on the other hand, emerge as the primary land cover type influencing the GTWS-CGR correlation (g = 37%), despite only covering 10% of the land surface. Semi-arid regions also play an important part, contributing a substantial fraction to both the GTWS signal (f = 25%) and

255

the GTWS-CGR correlation (g = 26%). However, semi-arid regions cover approximately 25% of the land surface area, a much larger portion than tropical forests, highlighting the much stronger role of tropical forests in governing the GTWS-CGR relationship per unit area.

Figure 4 presents a time series of the regional TWS variations through the whole period 2002-2023. Figure 4a regionalises across the five spatial regions shown in Figure 3a. Each region is accounted for such that the sum of positive and negative contributions equals the GTWS, in black, while the CGR variations are also shown against a separate axis. While the NH extratropics show large TWS variations this often tends to be anti-correlated with TWS elsewhere, and there is little correlation with the CGR. By contrast there is a particularly consistent alignment between the timing of many CGR events and the variations in TWS across tropical America. After 2020 the CGR and TWS signals both become more variable without the clear peaks or troughs seen earlier, although there is still reasonable alignment of the GTWS and CGR signals up to the end of 2022. Overall, this figure demonstrates that tropical regions generally contribute significantly to the GTWS signal and are consistently coherent with CGR variations. Whereas, even substantial variations in the NH extratropics, where there is a lot of anti-correlation with the GTWS, are thus unlikely to be significantly impacting the CGR. This pattern is also true for the SH extratropics, although the magnitude of variations is smaller here than in the NH.

Now we consider regionalising the IAV in GRACE TWS and its correlations with the CGR, according to land cover types, using the MODIS land cover map (Supplementary Figure 1). Figure 4b shows the contribution to the GTWS IAV for each land cover type plotted against the right (inverted) axis, with the CGR on the left axis. The largest CGR peak due to the 2015/2016 El Niño event very closely corresponds to the water storage variations seen in tropical forests, where there is a substantial TWS minimum over this period. The CGR returned to reference levels in 2017, and global TWS recovered shortly afterwards. Note that we might also associate the increase in CGR around 2005 with the tropical forests, although this event is not noticeable in the global storage signal. This increase coincides with the 2005 drought in the Amazon driven, not by El Niño, but by elevated tropical North Atlantic sea surface temperatures, affecting the southern two-thirds of Amazonia (Phillips et al., 2009). Then in 2009, a notable drop in CGR aligns with a peak in tropical forest TWS, an occurrence associated with the exceptional flood season observed in the lower Amazon basin during the first half of 2009, supposedly linked to the 2008–2009 La Niña event (Chen et al., 2010). This is followed by a sharp peak in CGR which coincides with another drought of 2010 in Amazonia, more severe than in 2005. This event started during El Niño and was then intensified as a consequence of the warming of the tropical North Atlantic (Marengo et al., 2011).

A decrease in CGR can also be observed in 2011, indicative of a strong land carbon sink. This sink anomaly has been corroborated by independent observations from MODIS EVI and GOME-2 SIF showing much enhanced photosynthetic activity over this period (Ma et al., 2016), as well as by GOSAT XCO<sub>2</sub> (Detmers et al., 2015). This feature relates to an increase in TWS in semi-arid regions, primarily resulting from a wet period in Australia during 2010-2011. Large parts of semi-arid Australia are covered by endorheic basins, where TWS variations are highly sensitive to precipitation (Petch et al., 2023b). Our results align with Poulter et al. (2014), who found that nearly 60% of carbon uptake over this period could be attributed to semi-arid Australian ecosystems, which experienced several consecutive seasons of increased precipitation due to prevalent La Niña conditions. However, other features in the CGR variability are less clearly associated with TWS variations partitioned

**Figure 4.** GRACE terrestrial water storage (TWS) interannual variability and global CO<sub>2</sub> growth rate (CGR) 2002-2023. (a) aggregated by continental regions (b) by land cover type.

in this way. In 2020, the CGR exhibited an opposite response compared to what we normally observe in relation to the tropical forest TWS anomaly. The global TWS anomaly remained positive, possibly due to conditions in other regions (notably wet in NH extratropics and Tropical Africa from Figure 4a). This could also be an indication of a lack of TWS influence on the CGR and may be caused by external factors, such as reduced human emissions due to the COVID-19 pandemic.

A clear difference between the Figures 4a,b TWS signals is that the vegetation cover contributions tend to be more aligned with the global TWS signal without the strong cancellations seen in continental TWS. This is largely explained by croplands and grasslands which often have large compensating TWS anomalies distributed between the NH extratropics and the tropical zones on all continents. This can be demonstrated by the disappearance of the pronounced anomalies depicted in Figure 4a when croplands and grasslands are excluded from the analysis (see supplementary Figures 2 and 3). This is an example of

compensatory effects of water anomalies discussed in Jung et al. (2017), which could lead us to underestimate the overall CGR role of croplands and grasslands.

## 4 Regional flux contributions to CGR estimated by atmospheric inverse methods

The previous section looked at regionalising TWS with respect to CGR correlations. In contrast this section will try to regionalise CGR contributions by using atmospheric CO<sub>2</sub> inversion model products. This approach offers the advantage of identifying the regions that directly contribute most to CGR variability without depending on the coupling between TWS and CGR. Additionally, these products potentially allow us to gain direct insight into the relative magnitude of the CO<sub>2</sub> flux contributions from each region, which were previously only inferred from regional TWS variations. The reliability of this approach will however be limited by the quality of inversion products. Therefore, we employ multiple products to assess their agreement with each other, as well as with the previous GRACE results, to provide further insights.

**Figure 5.** Interannual variability of observed atmospheric CO<sub>2</sub> growth rate (CGR) dervived from GML NOAA surface flasks (red dashed), and global terrestrial carbon flux interannual variability from an ensemble of eight atmospheric CO<sub>2</sub> inversion products over 2002-2022. Ensemble mean, black, with product range in grey.

Figure 5 compares the global land surface CO<sub>2</sub> flux created from the ensemble of eight different inversion products, and the CGR interannual variability derived from surface flasks. The shaded region represents the maximum and minimum across the products around the mean. The products, when integrated globally, show good agreement with each other and with the surface flask based CGR in terms of correlation, although there are substantial variations in the amplitude of the resulting terrestrial IAV. These results underscore that the majority of the IAV originates from the land, with minimal contributions coming from variations in ocean uptake and fossil fuel emissions. However, there is reduced agreement between the CGR and the inversion products between 2020 and 2021, e.g. CGR observations and inversion anomalies may be of opposite sign. The low CGR may result from anthropogenic emissions being 8.8% lower in the first half of 2020 compared to the same period in the previous

year, as a result of the COVID-19 pandemic impacting human activities (Liu et al., 2020). In this case pure detrending is unlikely to remove the anthropogenic signal in the CGR, which could explain this result.

In order to compare the inversion products at finer spatial scales, Figure 6 shows the contribution of each land grid point to the global land surface flux IAV, from equation (1), for each inversion product over the period January 2002 to December 2022. While there are some similarities on larger scales (see below), the maps also highlight many differences between the products. Such discrepancies will limit the value of these products for fine scale analysis.

The aggregated regional  $CO_2$  contributions for each inversion product, expressed as a percentage of the global CGR, are shown in Figure 7. The ensemble means show agreement with GRACE in attributing most of the terrestrial carbon variability to the tropical regions (f = 74%), based on equation (1). NISMON- $CO_2$  strongly favours tropical America, while CAMS and UoE show tropical Africa as the dominant contributor. IAPCAS and MIROC largely favour the NH extratropics. We will quantify this spread among products later.

There is general consensus among the products regarding contributions from land cover types, Figure 7b, with all products attributing nearly all CGR variability to three vegetation types: semi-arid areas, tropical forests, and grasslands/croplands. The ensemble means highlight semi-arid regions (f = 38%) and tropical forests (f = 35%) as the primary contributors, aligning with the GRACE data, which identified these two regions as the most dominant contributors to the GTWS-CGR correlation in Figure 3b.

However, the NISMON-CO<sub>2</sub> product stands as an outlier, assigning a particularly large contribution to tropical forests. A contributing factor may also be the use of the NICAM transport model, which has previously been shown to yield distinct flux patterns compared to other models, especially in the tropics (Peylin et al., 2013). For instance, NICAM has produced stronger carbon release signals and broader IAV during major El Niño events such as 1997/1998. Additionally, sparse observational constraints in tropical regions may exacerbate differences between inversion systems, allowing model structure and transport errors to exert a strong influence. Nevertheless, this observational limitation is common across systems and is therefore unlikely to be the sole driver of NISMON's divergence. A more targeted model intercomparison would be necessary to isolate the specific causes of this discrepancy.

We also find that posterior estimates for large regions exhibit IAV that is not present in the a-priori biosphere flux estimates, in the products for which priors were available. In many cases the priors include only seasonal cycles and lack substantial interannual variation, indicating that the IAV in the posterior fluxes is not inherited from the prior. This means that observational constraints and transport model dynamics play a key role in shaping the posterior variability.

Interestingly, there is much less consensus regarding spatial region contributions from Figure 7a. For example, IAPCAS shows minimal contributions from tropical America, while contributions from the northern extratropics vary widely across products. IAPCAS and CTE attribute around 40% of CGR variability to the NH extratropics, whereas CAMS attributes only 12%. This spread reflects a known challenge in attributing carbon variability to the NH extratropics, particularly when relying on surface observation inversions. Previous studies have highlighted that inversions using only in situ CO<sub>2</sub> data may underestimate variability in these regions. Guerlet et al. (2013) for example, suggest that satellite observations show more substantial flux variability in the NH extratropics than is captured by surface-only inversions.

Figure 6. Maps of  $1^{\circ} \times 1^{\circ}$  grid point contributions to global land carbon flux IAV for eight atmospheric CO<sub>2</sub> inversion products.

Figure 8 presents time series illustrating how different regions contribute to the total land  $CO_2$  flux variability, based on the ensemble mean of the inversion products, with the CGR line shown in red. This allows us to see how different regions

Figure 7. Regional contributions to total terrestrial  $CO_2$  flux IAV for eight different atmospheric  $CO_2$  inversion products and ensemble mean over 2002-2022. (a) shows continental regions while (b) shows vegetation cover type regions.

may serve as dominant drivers of CGR variability at different times throughout the 20-year period, similar to Figure 4. The colour bar inserts on Figure 8 represent the standard deviations of the eight inversion products around the ensemble mean for the whole 20-year period, sharing the same scale as the main axis used in the figure. The standard deviations for continental regions average 0.32 GtC yr<sup>-1</sup>, whereas for land cover types the average is significantly lower, 0.21 GtC yr<sup>-1</sup>. We also examine how the contribution from each land cover type varies during different temporal CGR events, in Figure 8b.

During the period of elevated CGR levels around 2002 (reduced uptake by the land), similar contributions are observed from semi-arid regions and croplands and grasslands, with some input from tropical forests. This peak is thought to possibly be

**Figure 8.** Contribution to terrestrial IAV from atmospheric CO<sub>2</sub> inversion products 2002-2022. (a) aggregated by continental regions, (b) by land cover type.

associated with Northern Hemisphere fires (Jones and Cox, 2005). The decrease in CGR observed in 2004 can primarily be attributed to cropland and grasslands according to the inversion products, although the timing of this event does not exhibit a strong correlation with the GRACE TWS variations in Figure 4b, suggesting that water availability may not be a key driver. In 2005 tropical forests and semi-arid regions emerge as the dominant contributors to the positive CGR anomaly, which agrees well with the GRACE data showing droughts across these regions, despite there being a positive global TWS anomaly during this period. The inversion products also align with GRACE in attributing the 2008-2009 decrease in CGR mostly to tropical forests and the 2011 decrease mostly to semi-arid regions. During the substantial increase in CGR coinciding with the 2015/16

El Niño the inversion products show comparable contributions from semi-arid regions, croplands and grasslands and tropical forest, whereas the GRACE data most strongly highlight the role of tropical forests, illustrated in Figure 4b. Overall the inversion products show minimal CGR IAV contributions from sparsely vegetated regions, and little from tundra and Arctic shrublands and extratropical forests, also in agreement with the GRACE GTWS-CGR correlation contributions, Figure 3b.

From Figure 8a it is notable that the NH extratropics exhibits considerably less CO<sub>2</sub> flux variability compared to the TWS variability, which is consistent with the insignificant contribution to the GTWS-CGR correlation. The overall contribution to the global land IAV appears more uniformly distributed across spatial regions compared to land cover types, suggesting that factors related to land cover type may play a more significant role in determining the terrestrial carbon flux variability than spatial regions *per se*. Examining specific time periods reveals interesting regional patterns. For instance, during 2002-2003, the inversion products indicate a period of low uptake from the land, leading to high CGR, primarily driven by variations in the tropics, particularly tropical Africa, despite GRACE observations showing positive tropical TWS variations at this time, Figure 4. Atmospheric data for the inverse models in these regions were sparse until around 2010, which could be a possible cause of this discrepancy.

The drop in CGR during 2004 appears to coincide with increased land uptake in tropical Eurasia and Australia, and tropical Africa, again with limited evidence of TWS influence in these regions. The large CGR drops in 2009 and 2014 appear to be predominately driven by the NH extratropics, despite GRACE only indicating possible contributions from tropical America with minimal input from the NH. In 2022, tropical America stands out as a significant influence for the decline in CGR in the inversion products, which is also accompanied by a clear increase in TWS in the same region. From Figure 8a it is notable that the NH extratropics exhibit considerably less CO<sub>2</sub> flux variability compared to the TWS variability, which is consistent with the insignificant contribution to the GTWS-CGR correlation. Since the larger continental spreads seen in Figure 4a are not reproduced Figure 8a, this suggests that the inverse models do not generally capture the same anti-correlated behaviour seen in water anomalies. This could be because the CO<sub>2</sub> flux IAV occurs only at larger spatial scales or because the atmospheric inversion models lack the spatial resolution to detect smaller scale anti-correlated fluxes. We now perform a sensitivity analysis to try further bring out regional relations.

We calculate local sensitivities of the terrestrial CO<sub>2</sub> flux to GRACE TWS interannual variations using the inversion products. This provides insight into how changes in TWS translate into carbon flux variations and can help us relate Figure 4 to Figure 8. Sensitivities allow us to assess how water-limited or energy-limited conditions might modulate the influence of TWS anomalies on NBP. For instance, a small TWS anomaly in a water-limited region may have a disproportionately large impact on CGR, while large TWS changes in energy-limited regions may have a smaller effect. Figure 9 shows the sensitivities calculated using the ensemble mean of the inversions for (a) spatial regions and (b) land cover type. The sensitivity is calculated using linear regression. These sensitivities do resemble the contributions seen in Figure 3. Figure 9a shows higher sensitivities in the tropics and smaller (insignificant) opposite sensitivities in the NH extratropics and SH extratropics. Figure 9b indicates strong CO<sub>2</sub> sensitivity to TWS variations in semi-arid regions, as well as high sensitivity in tropical forests, and croplands and grasslands. The high sensitivity in tropical forests indicates that TWS variations in these regions directly influence NBP, rather

than merely being correlated with CGR through external factors like ENSO. This reinforces our conclusion that tropical forests play a crucial role in regulating the CGR.

Overall, the total sensitivity is greater when aggregating by land cover type than by spatial regions. This suggests that aggregation by land cover better captures the variation in ecosystem responses to water availability, as different land cover types have distinct physiological and ecological responses to TWS anomalies. Aggregation by spatial regions can then obscure these differences, as regions often contain a mix of land cover types with varying sensitivities to water availability. For example, areas of high sensitivity to water availability in parts of tropical America (e.g tropical forests) can be masked when grouped into larger spatial regions that include ecosystems that are more sensitive to energy and less sensitive to water.

The strong correlation and sensitivity in tropical forests could be attributed to several possible factors. Tropical rainforest trees tend to have relatively shallow rooting systems and are hence more likely to be affected by changes in TWS when prolonged or severe droughts deplete soil moisture (Kleidon, 2004). While some trees can develop deep roots that provide drought resilience, the majority of water uptake in tropical forests occurs from shallower soil layers, making them especially sensitive to reductions in available water. Additionally, tropical forests generally have high water-use-efficiency (WUE), which allows them to maximize carbon uptake under favorable conditions but also makes them vulnerable to rapid declines in productivity when water becomes limiting (Keenan et al., 2013; Saleska et al., 2016). Another reason for this sensitivity could be due to the ongoing drying of parts of the Amazon—especially the southeastern region— which has pushed these forests into a state of heightened vulnerability to drought (Barkhordarian et al., 2019), further amplifying their sensitivity to interannual variability in water storage and the observed coupling between terrestrial CO<sub>2</sub> and water storage variability.

Figure 9. Regional  $CO_2$  flux sensitivity to TWS variations (units: kgC yr<sup>-1</sup>m<sup>-3</sup>), calculated using the ensemble mean of inversions. (a) Sensitivity by spatial regions, and (b) sensitivity by land cover type.

## 5 Discussion

The significant negative correlation observed between GTWS and CGR suggests a compelling link between water availability and terrestrial carbon uptake, where drier years coincide with elevated CGR levels, implying a weakening of the land carbon sink. This observation supports the notion that water availability plays a pivotal role as a limiting factor influencing land carbon uptake on interannual timescales. From Figure 4, it is also clear that even when the GTWS signal does not align with the CGR, regional TWS component signals sometimes closely follow the CGR, suggesting that water may be a key factor influencing CO<sub>2</sub> variations. The evidence for water scarcity restricting land uptake is clear during times of extreme drought. For instance, the Amazon which typically acts as an important carbon sink, experienced major droughts in the years 2005 and 2010 marked by notable peaks in the CGR. Additionally, the inversion products provide regional evidence supporting the impact of water availability during these events. However, it is important to note that other factors, including light availability and temperature, may also impact the interannual variability in land carbon uptake. This becomes apparent when regional variations in water availability do not consistently align with regional carbon variations from inversion products, suggesting weaker water-carbon coupling.

During the 2016 El Niño event, the high CGR was largely associated with reduced land uptake in tropical Africa, despite this region showing less dominant TWS variations. In this case, enhanced fire activity, reduced cloud cover (and thus increased radiation), and shifts in vegetation phenology have all been suggested as contributing factors to reduced carbon uptake (Liu et al., 2017; Zhu et al., 2018).

Our sensitivity analysis with the inverse model products indicated that the CO<sub>2</sub> flux was not highly sensitive to water variations in this region, suggesting that the variations in carbon uptake in tropical Africa were more likely driven by other factors beyond water availability during this event. Overall, our results demonstrate that water availability exerts a notable influence on interannual land carbon uptake. However, this impact exhibits spatial variability and is subject to influence from other environmental drivers which are not independent from water availability, adding complexity to the carbon-water relationship.

Given that tropical Africa shows a strong control on the CGR but low correlation with TWS, the value in understanding the relationship between CGR and TWS will primarily be in diagnosing the regional carbon cycle for areas such as the Amazon, for which numerous studies have shown evidence of it being vulnerable to water-related tipping points (Cox et al., 2000, e.g). The Amazon generates about half of its own rainfall (Salati et al., 1979) and so drought conditions have the potential to be amplified, thus creating a stronger carbon signal than would be observed otherwise. However, even in regions where the TWS-CGR relationship is not strong, knowing this helps deepen our understanding of which processes control the carbon cycle in that region and where modelling efforts can then be focused. An application of this technique will be to observe changes to the strength of the water control of the terrestrial carbon cycle with time, to understand how the processes are changing with shifts in climate.

We explored both land cover types and spatial regions because each offers a unique perspective. Examining different spatial regions allowed us to uncover regional patterns and variations, while considering land cover type can help highlight the role of vegetation in shaping carbon-water dynamics. We found that tropical forests contribute the most to the GTWS-CGR correlation

while covering only a small fraction of the Earth's land surface relative to other land cover types. Among tropical forests, the Amazon basin stands out from Figure 2, contributing a disproportionately large amount to the GTWS-CGR correlation (g = 28%) relative to its spatial coverage (< 5%). However, these are not the regions dominating the GTWS signal, nor are they the only regions controlling the land carbon uptake IAV. We found that semi-arid regions and croplands and grasslands contribute more to the GTWS signal, which is mainly due to their larger spatial coverage. This is consistent with Humphrey et al. (2018) who also looked at the TRENDY model results (Sitch et al., 2015), and FluxCOM datasets (Tramontana et al., 2016), as well as GRACE, to assess contributions. Ahlström et al. (2015) also found that semi-arid regions dominate the land carbon sink variability by calculating the contribution index using a biogeochemical dynamic global vegetation model (DGVM) LPJ-GUESS. They also compared results with an ensemble of TRENDY models and found similar partitioning.

The contribution to the GTWS-CGR correlation metric (equation 3) provides further insights that examining the TWS variability alone (equation 1) does not. For instance, the strong TWS variability signals seen in the NH extratropics or croplands and grasslands, only have small contributions to the global TWS correlation with CGR. However temporal filtering used in our TWS-CGR analysis may still be underestimating the true impact of water availability on CO<sub>2</sub> fluxes in the extratropics, which will be strongly seasonally sensitive (Keppel-Aleks et al., 2014).

Similarly, focusing solely on the regional TWS-CGR correlations does not provide information about the magnitude of the contributions to the global correlation. Therefore, integrating these three metrics provides a comprehensive analysis, which taken together more strongly indicate the role of tropical forests. This complements the work of Humphrey et al. (2018) who indicated a possible role for tropical forests due to the high correlation with CGR. However all three metrics only consider time-integrated information. The regional time series displays (Figure 4 for TWS, and Figure 8 for regional CO<sub>2</sub> fluxes) allow a more detailed analysis of particular events. Our results underscore the influence of tropical America along with tropical forests through the 20+ year period considered (Figure 3), however, it is also evident from both GRACE and the inversion products, that the dominant region influencing CGR IAV varies across different temporal events (Figure 4, 8). This is consistent with previous studies that have focused on individual events (e.g., Chen et al., 2010; Ma et al., 2016).

Our results show agreement with previous studies that have employed Earth System models (ESM) to assess CGR IAV. For instance, the dominant role of tropical America is also recognised by Martín-Gómez et al. (2023), who analysed various ESM simulations from CMIP6 over 1986-2013. They find that tropical regions, particularly tropical South America, tropical southern Africa, Southeast Asia, and parts of Oceania, contribute the most to interannual variations in CGR, with average land variance explained percentages of 22%, 10%, 5%, and 4% respectively. Additionally, Kim et al. (2016) analysed how interannual CGR is associated with ENSO using ESM simulations from CMIP5. During El Niño, strong NBP anomalies appear over most tropical land regions, particularly over Amazonia, Australia, and South Asia, and the Maritime Continent, where tropical rain forests exist. This is consistent with the regions we found to show high TWS-CGR correlations in Figure 2.

The process of aggregating local data into regional or global scales introduces challenges due to the complex spatial scale interactions. For example, Jung et al. (2017) demonstrated that local-to-global aggregation can result in different conclusions regarding the dominant driver of NEE. This was attributed to compensatory effects, where soil-moisture-controlled NEE anomalies showed strong spatial anti-correlation, leading to spatial compensation of positive and negative values. This meant

that when aggregated globally, temperature emerged at the dominant driver of NEE because its variability is on larger spatial scales, so the effects do not average out. Therefore, when interpreting results such as Figure 3, it is important to consider the potential role of spatial covariance.

We note also that the reported correlations between TWS and CGR have not been adjusted for co-varying climate factors such as temperature or VPD. As a result, these relationships may partly reflect indirect effects mediated by such variables. Covariance among climatic drivers—particularly between temperature, VPD, and water availability—introduces a degree of multicollinearity that complicates attribution of individual effects. For example, Humphrey et al. (2021) highlighted the difficulty of disentangling the respective roles of VPD and temperature on NBP variability due to their interrelated nature and showed that soil moisture impacts on NBP are shaped not only by direct water limitation but also by indirect feedbacks involving temperature and VPD. However, Humphrey et al. (2018) also found that most of the explanatory power contained in TWS could not be accounted for by temperature alone. Their partial correlation analysis showed that the CGR–TWS relationship remained significant even after controlling for temperature, suggesting that TWS captures a distinct and meaningful component of carbon cycle variability. Thus, while water availability cannot be interpreted in isolation from other variables, it still provides uniquely valuable information about ecosystem–climate interactions. Our findings should therefore be viewed as complementary to other studies that emphasise the roles of temperature or VPD.

We therefore turned to direct estimates of carbon fluxes from atmospheric inversion models which, while designed to constrain the atmospheric carbon budget, are often employed to regionalise surface carbon fluxes and investigate temporal variations, based on available atmospheric observations. Our analysis of eight such products demonstrate the consistency of their results when considering large scales, such as over continental regions or integrated over land cover types. However, limitations become apparent at smaller scales (Figure 6). As highlighted by Marcolla et al. (2017), fine-scale estimates of inversions are subject to limitations because atmospheric data may only effectively constrain larger-scale patterns comparable to the distances between monitoring stations. Globally, products agree because observationally the CGR is well constrained, set broadly by the difference between the slowly varying anthropogenic fossil fuel input and the well-measured accumulation rate of CO<sub>2</sub> in the atmosphere (Baker et al., 2006). However, the inversion models introduce factors contributing to discrepancies, such as the selection of the atmospheric CO<sub>2</sub> data, prior fluxes, and the choice of transport model. For example, Schuh et al. (2019) demonstrate that the transport models TM5 and GEOS-Chem, used by NOAA CT and UoE respectively, lead to systematic space-time differences in modelled distributions of CO<sub>2</sub>. Bastos et al. (2020) also found that at regional scales, differences between inversion products contribute the most to uncertainty in regional carbon budgets, whereas differences in DGVMs dominate uncertainty at the global scale. They emphasised that reducing the uncertainty in atmospheric inversions, for example through more observations in the tropics or the use of satellite-observations, is essential to reduce uncertainty in carbon budgets.

Here using the ensemble mean does offer a more representative picture by incorporating a range of estimates from different models, although different atmospheric inversion products may still be related through the same transport models or meteorological fields, allowing biases to remain. It is important that each individual ensemble member is well evaluated; otherwise, outliers or individual members that do not reproduce data can cause the ensemble mean to be misleading. Overall, our results

emphasise the need to improve consistency among inverse models at finer scales. This would be essential for achieving more reliable evidence of regional contributions to carbon cycle dynamics. In future work it would be beneficial to conduct a more comprehensive comparison of atmospheric CO<sub>2</sub> inversion products involving additional products beyond those considered in our study, e.g. containing satellite CO<sub>2</sub> observations.

#### 530 6 Conclusion

This study first assesses the relationship between TWS variations at global and regional scales with the interannual CGR over 2002-2022. This builds on the work of (Humphrey et al., 2018) by extending the TWS-CGR relationship up until 2023. The correlation between global TWS and CGR was r = -0.70 over the entire period, with a correlation of r = -0.69 observed for the period from 2002 to 2016, and r = -0.74 from 2016 to 2023. We expand on Humphrey et al.'s analysis by separately mapping regional TWS-CGR correlations and their relative contributions to the GTWS-CGR correlation. This allows for better identification of key regions where water limitation is likely to be influencing surface  $CO_2$  fluxes and thus contributing to the GTWS-CGR correlation.

Our analysis reveals that the tropics alone can explain the entire GTWS-CGR correlation. Specifically, tropical America emerges as the largest contributing area, accounting for g = 69% of the GTWS-CGR correlation, despite only representing f = 40% of the GTWS variance and covering less than 12% of the Earth's land surface area. Moreover, we observe minor cancelling contributions to the GTWS-CGR correlation from the Northern (g = +17%) and Southern (g = -15%) Hemisphere extratropics, which represent f = 38% and f = -12% of GTWS variability signal respectively.

Tropical forests emerge as the primary land cover type influencing the global correlation (g = 37%), despite being less important for the global TWS signal (f = 22%). Semi-arid regions also play an important role in contributing to both the GTWS-CGR correlation and the GTWS signal (g = 26%, f = 23%). Notably, tropical forests cover only 10% of the land surface area, hence they exert a disproportionately stronger impact relative to their spatial extent compared to semi-arid regions, which encompass 25% of the land surface. Croplands and grasslands contribute largely to GTWS variability (f = 22%) due to large spatial coverage but have a relatively low correlation with the CGR (r = -0.27) and consequently contribute less strongly to the GTWS-CGR correlations (g = 15%).

We also employed eight different atmospheric  $CO_2$  transport inversion model products to directly assess the regional contributions to CGR interannual variability. The ensemble mean of the eight products shows good agreement with the CGR globally, and these products also demonstrate considerable consistency among themselves on large scales, however their agreement diminishes when examined at finer spatial scales. In alignment with GRACE TWS data, the ensemble mean of the inversion products also attribute the majority of terrestrial carbon interannual variability to the tropics (f = 74%), with smaller contributions from the NH extratropics (f = 15%) as well as a small (f = 11%) positive contribution from the SH extratropics, in contrast to GRACE. The inversion products all show broad agreement with GRACE in the dominant land cover types con-

tributing to the CGR variability. Although not strongly trusting the regional inverse products we also calculated regional-scale sensitivities of inverse model NBP against TWS, and find consistency with the TWS-CGR correlation contribution results.

Finally for both water storage variability and inverse model derived CO<sub>2</sub> fluxes we look at the interannual variation time series and note that the dominant anomaly regions do change over the 20-year period, and we discuss how this ties in with literature on patterns of drought over time. These time series also reveal cases where regional TWS signals often closely follow the CGR even when the global TWS does not, thus providing further evidence of the importance of water availability as a constraint on regional CO<sub>2</sub> fluxes. Overall, our results suggest that terrestrial ecosystem models should prioritize improved representation of water constraints, particularly in tropical forests where water availability plays a critical role in modulating carbon fluxes.

Data availability. The MODIS land cover data is available from https://lpdaac.usgs.gov. The CGR data Lan et al. (2016) was retrieved from https://gml.noaa.gov/ccgg/trends/gl\_data.html. The GRACE TWS data can be obtained from http://grace.jpl.nasa.gov/data/get-data/jpl\_global\_mascons/. All the atmospheric CO<sub>2</sub> inversion products are publicly available from

570 https://meta.icos-cp.eu/objects/FHbD8OTgCb7Tlvs991UDApO0.

Author contributions. This work was carried out by SP as part of her PhD. All the analysis and figures were generated by her as well as a first draft of all the paper text. KH, TQ and RK acted as PhD supervisors. LF and PP provided data resources. Each contributed edits to the paper text.

Competing interests. The authors declare that they have no conflict of interest

Acknowledgements. This research has been supported by the SCENARIO NERC DTP programme (grant no. NE/S007261/1) and from a CASE award from the UK Met Office for SP. KH and TQ would also like to acknowledge the support of the NERC NCEO International Programme (grant no. NE/X006328/1). LF and PIP received support from the UK Space Agency and the UK National Centre for Earth Observation funded by the National Environment Research Council (grant no. NE/R016518/1).

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
