# Peer review of "Strong relation between atmospheric CO2 growth rate and terrestrial water storage in tropical forests on interannual timescales"

_EGUsphere, 2025_

## Author Comment (AC1)

**The authors of "Strong relation between atmospheric CO2 growth rate and terrestrial water storage in tropical forests on interannual timescales" have provided a very thorough and comprehensive analysis of interannual variability of terrestrial water storage and its link to the atmospheric growth rate of CO2. This research adds to our knowledge of interannual variability of land carbon fluxes and opens new pathways for discovery in this direction. The results are interesting and the research should be published with some modifications. However, I would really implore the authors to deepen their discussion, which I will elaborate below. Additionally, I urge the authors to use a broader spectrum of inverse models. Some other, smaller comments are provided after.**

We would like to thank the reviewer for their thorough and thoughtful feedback. We greatly appreciate the time and effort taken to review our manuscript.  Below, we provide a response to each comment outlining our proposed revisions and clarifications.

**General comments**

**Throughout the discussion, the authors discuss the findings, but fail to provide some deeper context. In my opinion, the interesting findings of this manuscript should be placed in context better. This would improve the impact of this manuscript, as it allows for follow-up research. I would urge the authors to focus more on the (possible) mechanisms that drive the findings in this manuscript.**
**For example, the authors do not mention why the tropics account for such a large portion of the covariance. Is this because the IAV in the tropics outweighs IAV in temperate regions (not according to the inversions)? Or is this because droughts in the tropics (i.e. the ENSO cycle) covers the entire tropics (whereas droughts in temperate regions are more driven by synoptic variability and thus happen over smaller scales). Additionally, the authors should mention why TWS is a better explanatory factor than e.g. VPD or temperature. In L.616, the covariance between temperature and water availability is mentioned but not discussed sufficiently.**

**Finally, the positive correlation between temperate TWS and carbon growth rate should be mentioned, regardless of the small contribution to the growth rate. Can this be explained physically?**

We appreciate this comment and agree that providing deeper context and mechanistic insight will significantly strengthen the discussion. We believe the tropics account for a large portion of variability due to a combination of spatial coherence in the tropics and large amounts of IAV. These two things can be difficult to disentangle. In the revised manuscript we will include deeper discussion of this and refer to additional references which also highlight the importance of the tropics. We will include text such as:

- The tropics play a disproportionately large role in driving the covariance between CGR and regional TWS. This is due to a combination of factors: (i) the tropics exhibit some of the

largest magnitudes of IAV (ii) Climate anomalies such as ENSO events often synchronise drought and temperature anomalies across vast tropical regions, creating spatially coherent signals that are amplified at the global scale. In contrast, temperate regions typically experience more localized and less synchronised climate variability, resulting in smaller contributions to global CGR variance.

I don't think we can say for certain that TWS shows more explanatory power than VPD, but we will include discussion of this and highlight potential benefits of using TWS, and our rationale for our focus on TWS and the potential advantages it offers. One key motivation is the availability of large-scale observational data from the GRACE satellites. In contrast, at large scales VPD often relies on model outputs; for example, He et al (2022), demonstrated the significant negative correlation between VPD and NEP IAV using FluxCOM and TRENDY data, whereas our study emphasises an observational approach. Additionally, TWS reflects integrated water availability across all components of the hydrological cycle, while VPD primarily represents atmospheric demand. TWS can capture the cumulative effects of prolonged droughts on ecosystem productivity, which is particularly relevant for understanding carbon cycle dynamics. Although we do not claim that TWS is a better explanatory variable to temperature or VPD, it offers a complementary perspective and allows us to leverage independent observational datasets.

We will include more in depth discussion about covariance between temperature and water e.g highlighting methods Humphrey et al (2018) used to demonstrate the effects of water on CO2 variability held independent of temperature effects.

Additionally, we will discuss the covariance between temperature and water availability more thoroughly, including how multicollinearity may complicate interpretations. We will comment on the observed positive correlation between temperate TWS and carbon growth rate, particularly in regions like South Brazil and East China, and provide possible biophysical explanations such as these regions having an opposite response to ENSO.

**Additionally, the analyses should be done with more than four inverse models. The GCB inverse fluxes are made publicly available (https://meta.icos-cp.eu/objects/FHbD8OTgCb7Tlvs99lUDApO0 for GCB2023 and more recently https://meta.icos-cp.eu/objects/GpFcABoKcZMVnRUlLHRInhdM for GCB2024). Some of these models indeed only cover the OCO2 period (2015 onwards), but for GCB2023 and 2024, 8 systems with sufficient data are provided. Therefore, I expect the analyses to be done with all available models.**

We agree with the reviewer's suggestion that additional inversion products will strengthen our analysis. We will incorporate all inversion models made available through GCB2023 that have suitable temporal availability. The updated analysis will include CarboScope, CTE, IAPCAS and MIROC models along with the existing products used (CAMS, UOE, NISMON_CO2 and NOAA CT). This will involve updating Figures 6,7,8,9 and 10 and necessary text to be consistent with new results. Below shows updated Figures 7 and 8 with additional inversion products, Figures 6,9 and 10 look very similar to before and main conclusions have not changed.

[Figure]

[Figure]

**Technical comments**

**1. L.75: A reader might find it strange that the first sentence of the plain language summary says the CO2 increases every year, but reads here that there is a decrease (which I understand is in the growth rate, so there could still be an increase). I would recommend to maybe rephrase to larger and smaller growth rates with El Nino/La Nina**

Thank you for highlighting this. We will revise the wording to clarify that while atmospheric CO2 levels are increasing every year the increase in some years is smaller than other years i.e larger during El Niño years and smaller during La Niña years.

**2. L.96: "Most vegetation responds to soil moisture". Yes, but also to VPD (which is not included in TWS). These effects are difficult (if not impossible) to disentangle, but interesting to mention.**

Thank you for pointing this out, we very much agree with this point. We will mention the challenges in disentangling these effects due to the fundamental issue that VPD and soil moisture are coupled, where high VPD usually corresponds to dry soils and low VPD to wet soil. We will include references such as He et al (2022) which explores the impact of VPD on CO2 variability.

**3**. **L.106: I think your definition of NBE is the same as the definition of NEE used here. It could be easier for a general reader to use only one of the two.**

Will change text to be consistent throughout and use NEE.

**4. L.237: Linear detrending removes the fossil fuel trend, but also any trend in biogenic and ocean fluxes. This should be mentioned**

We will clarify that while linear detrending is used to remove the long-term fossil fuel-driven trend in CO2, it also removes any long-term trends in biogenic and ocean fluxes.

**5. L.238: Why are the CGR and TWS smoothed using different windows? To me it makes intuitive sense to use the same window**

We chose these smoothing windows to be consistent with previous literature i.e Humphrey et al. (2018) who also chose different windows to give best readability of figures. The different smoothing windows were used because the CGR data is inherently noisier than TWS and required slightly stronger smoothing to reveal meaningful interannual variability. Note that the mean seasonal cycle is in any case already removed before smoothing. We will add a sentence to clarify this in text.

6. **Fig 2: Here, the growth rate can become negative (so the first sentence of the introduction is not true, or is this detrended growth rate?)**

The growth rate has been detrended. We will clarify in text any confusion about what the negative growth rate means. By Figure 2 we have added "It is important to note that, although atmospheric $CO_2$ concentrations continue to increase each year, the CGR has been detrended; therefore, negative CGR anomalies reflect periods of below-average growth relative to the long-term mean growth rate."

7. **Fig 3: The positive correlations in south Brazil and east China are quite interesting, and could be explained biophysically. This links to my first general comment as well.**

The area in South America is dominated by grassland. There could be a number of reasons for this result. These regions are thought to have an opposite response to ENSO.. However, it could also be the case that these regions are not having a large impact on the global CGR. We will include discussion of this in text.

**8. L.305: It would be nice to add a horizontal line showing the global correlation to make it more explicit that the tropics can indeed explain all the correlation**

We will revise the figure to include a horizontal line indicating the global correlation value at -0.70. See updated figure below:

[Figure]

(a) GRACE TWS-CGR Spatial Regions Correlations and Contributions

(b) GRACE TWS-CGR Land Cover Correlations and Contributions

**9. L.432: It's interesting that NISMON has a larger range, and I recommend the authors to (shortly) discuss any potential reason for this (e.g. prior model, observations used, transport model). The inclusion of other models might help in this.**

The inclusion of additional inversion models in our analysis does indeed highlight that NISMON stands out for the particularly large contribution from tropical forests. While prior fluxes typically contribute relatively little to interannual variability, a potential factor influencing NISMON's larger range could be the transport model (NICAM). However, a more targeted analysis—such as that done in intercomparison projects like TRANSCOM (e.g., Baker et al., 2006)—would be needed to draw firmer conclusions. A lack of observational constraints in the tropics may also contribute to this variability; however, this limitation is common to many inversion systems and does not fully explain the observed differences.

Peylin et al. (2013) also found that NICAM differed from other inversion systems, particularly in the tropics, where it produced a larger carbon release and broader IAV response during major El Niño events such as 1997/1998.

**10. L.537: I'm not sure I understand the phrasing 'still a key factor'. Is this in contrast to other studies that pointed towards temperature? Otherwise just remove the 'still'**

Our intent was not to contrast with studies emphasizing temperature, but rather to highlight that—even though the global water storage signal does not always align with the global CGR—regional water storage anomalies (as shown in Figure 5) do exhibit interannual variations that coincide with CGR changes during certain periods. Therefore, water may still play an important role,

even if this is not immediately apparent from the global averages alone. We will remove the word 'still' to avoid confusion.

**11. L.624: I disagree with the statement that atmospheric inverse models are "specifically designed to regionalise carbon fluxes". Atmospheric inversions cannot distinguish CO2 from different regions, and mainly constrain the atmospheric carbon budget. Especially observation-sparse regions like the tropics are hard to constrain by atmospheric inversions (even satellite-based inversions). This does not mean that inversions cannot be used to analyse regional impacts -- I think your use of multiple models is suitable as it covers a range of flux realisations. And although I appreciate the downtuning later in this paragraph, I think this statement needs refinement.**

We will revise this sentence to say "Atmospheric inversion models  designed to constrain the atmospheric carbon budget, but are often employed to regionalise carbon fluxes and investigate temporal variations, based on available atmospheric observations."

References

Baker, D. F., Law, R. M., Gurney, K. R., Rayner, P. J., Peylin, P., Denning, A. S., ... & Yuen, C.-W. (2006). TransCom 3 inversion intercomparison: Impact of transport model errors on the interannual variability of regional $CO_2$ fluxes, 1988–2003. Global Biogeochemical Cycles, 20, GB1002. https://doi.org/10.1029/2004GB002439

He, B., Chen, C., Lin, S., Yuan, W., Chen, H. W., Chen, D., Zhang, Y., Guo, L., Zhao, X., Liu, X., Piao, S., Zhong, Z., Wang, R., & Tang, R. (2022). Worldwide impacts of atmospheric vapor pressure deficit on the interannual variability of terrestrial carbon sinks. National Science Review. https://doi.org/10.1093/nsr/nwab150

Humphrey, V., Zscheischler, J., Ciais, P., Seneviratne, S. I., Friedlingstein, P., Green, J. K., ... & Peters, W. (2018). Sensitivity of atmospheric $CO_2$ growth rate to observed changes in terrestrial water storage. Nature, 560(7720), 628–631. https://doi.org/10.1038/s41586-018-0424-4

Peylin, P., Law, R. M., Gurney, K. R., Chevallier, F., Jacobson, A. R., Maki, T., Niwa, Y., Patra, P. K., Peters, W., Rayner, P. J., Rödenbeck, C., van der Laan-Luijkx, I. T., & Zhang, X. (2013). Global atmospheric carbon budget: Results from an ensemble of atmospheric $CO_2$ inversions. Biogeosciences, 10, 6699–6720. https://doi.org/10.5194/bg-10-6699-2013

---

## Author Comment (AC2)

*Response to RC2:*

**This manuscript presents a comprehensive assessment of the relationship between atmospheric CO₂ growth rate (CGR) and terrestrial water storage (TWS) at interannual timescales, with a particular focus on tropical forest ecosystems. By combining satellite-derived GRACE data, multiple atmospheric CO₂ inversion products, and land cover datasets, the authors demonstrate a consistent and statistically significant negative correlation (r = -0.70) between global TWS and CGR from 2002 to 2023. Notably, tropical America and tropical forests emerge as dominant contributors to this coupling, despite their relatively small spatial extent. The study also applies multiple methods to partition the spatial and functional drivers of this relationship and evaluates the robustness of the signal using both observational and model-based approaches.**

We thank Reviewer 2 for their thoughtful and constructive comments. Below, we provide a point-by-point response, indicating how we will address each suggestion.

**Overall, this study is of high quality and reflects substantial analytical and conceptual work. The manuscript is well structured and clearly written. I have several comments and suggestions for the authors to consider:**

**The abstract reports a strong TWS–CGR correlation, but it lacks a statement explaining how this finding advances previous work (e.g., Humphrey et al., 2018). Please clarify further how this study uniquely extends or deepens our understanding.**

In the revised abstract, we will explicitly state how our study builds on and extends prior work, particularly by providing a more detailed regional contribution and temporal breakdown of contributions (e.g., Figures 5 and 9).

**Lines 28-31: The sentence "tropical forests exhibit the strongest CGR correlations" is important but could briefly explain why—e.g., due to high productivity sensitivity to water stress—so help readers understand the physiological context.**

We will revise the abstract to say "Tropical forests exhibit the strongest CGR correlations due to their high productivity and sensitivity to water stress, which strongly influence interannual variations in carbon uptake."

**In Section 2.3, clarify whether all four inversion products use harmonized fossil fuel and biomass burning emissions (e.g., GFED versions). Differences in fire emissions datasets could bias regional flux attribution.**

We have re-done the analysis using 8 inversions from GBC2023 that share harmonized fossil fuel and fire emissions (e.g., using the same GFED version). This does not change our conclusions and we will update the text accordingly and clarify this harmonization in Section 2.3.

**Table 1 summarizes inversion methods, but the main text should include 2–3 sentences interpreting key differences in transport models, meteorological fields, or prior flux assumptions and how they might affect tropical vs. extratropical estimates.**

We will add text such as
"Atmospheric $CO_2$ inversion products differ in their use of transport models, meteorological inputs, and prior flux assumptions—all of which can significantly influence flux estimates, particularly when comparing tropical and extratropical regions (Peylin et al., 2013; Chevallier et al., 2010). In the tropics, flux estimates are especially sensitive to how transport models represent deep convection and vertical mixing, while extratropical estimates are more influenced by synoptic-scale advection and boundary layer dynamics. Prior flux assumptions, such as prescribed seasonal cycles or vegetation responses, can also introduce regional biases—especially in the tropics where these assumptions often fail to capture complex climate–ecosystem interactions (Munassar et al., 2022; Gaubert et al., 2019). Moreover, the relative scarcity of $CO_2$ observations in tropical regions amplifies the impact of model structure and prior uncertainty, whereas denser observational networks in the extratropics provide stronger constraints on inversion results (Patra et al., 2005; Schuh et al., 2019).

**I don't think it's necessary to place Figure 1 in the main text. It is recommended to put it in the supplementary files. Overall, there are too many figures in the full text. It is suggested to combine them.**

We can move Figure 1 to the supplementary.

**Lines 293-294: Please explain why some regions with high local correlation contribute minimally to the global signal—e.g., due to small TWS variance or maybe small spatial extent—and explicitly state how these cases are handled.**

We will add a discussion clarifying that regions with high local correlations may contribute minimally to the global TWS–CGR correlation if they exhibit low TWS variability or have limited spatial extent. Conversely, other regions (e.g., the Northern Hemisphere extratropics) may show large TWS variability but still contribute little due to weak correlation with CGR. Overall, these features highlight the usefulness of considering the three metrics together.

**In Figure 4, report the number of grid cells per land cover class and provide standard deviations or interquartile ranges to contextualize variability in contribution estimates.**
We have included in Figure 1 caption the percentage of the land surface each region occupies. We can quote regional areas in text (area perhaps better as grid cells vary in size). We can also report standard deviations of storage variations in each region.

**Annotate key ENSO or drought years (e.g., 2005, 2010, 2015–16) directly in Figure 5 to aid interpretation of CGR–TWS relationships and align with the narrative.**

We have added shading in Figure 5 to indicate ENSO index using ONI index from
https://origin.cpc.ncep.noaa.gov/products/analysis_monitoring/ensostuff/ONI_v5.php

[Figure]

**lines 390–395: reference a figure or appendix that visualizes cross-regional TWS anomaly compensation (e.g., a correlation matrix or spatial covariance map), supporting the claim of cancellation effects in croplands.**

We will add supplementary figures which demonstrate the cancelling effects seen in the Northern Hemisphere appear to occur within croplands and grassland. See figures below, which show contributions of croplands and grassland in the NHet and tropics + SHet in top figure, then the bottom figure shows the same as Figure 5a but with croplands and grasslands removed from the NHet, where we no longer see the large storage variations that were causing the cancelling effect.

[Figure]

**The sensitivity analysis in Figure 10 is valuable, but the ecological interpretation of why tropical forests show both high correlation and high sensitivity should be more deeply discussed—e.g., in terms of water-use efficiency or rooting depth.**

We will expand the discussion on ecological mechanisms, including deeper rooting systems, water-use efficiency, and stomatal control, to explain both high correlation and sensitivity in tropical forests. We will add text such as

The strong correlation and sensitivity in tropical forests could be attributed to several possible factors. For instance, tropical rainforest trees tend to have relatively shallow rooting systems and are hence more likely to be affected by changes in TWS when prolonged or severe droughts deplete soil moisture (Kleidon, 2004). While some trees can develop deep roots that provide drought resilience, the majority of water uptake in tropical forests occurs from shallower soil layers, making them especially sensitive to reductions in available water.
Additionally, tropical forests generally have high WUE, which allows them to maximize carbon uptake under favorable conditions but also makes them vulnerable to rapid declines in productivity when water becomes limiting (Keenan et al., 2013; Saleska et al., 2016).  Another reason for this sensitivity could be due to the ongoing drying of parts of the Amazon—especially the southeastern region—which has pushed these forests into a state of heightened vulnerability to drought, further amplifying their sensitivity to interannual variability in water storage and the observed coupling between terrestrial $CO_2$ and water storage variability.

**lines 530–534: Clarify whether TWS–CGR correlations were adjusted for or confounded by co-varying climate factors (e.g., temperature, VPD). If not adjusted, include a cautionary note on potential indirect effects.**

We will add a cautionary note to say "We acknowledge that the reported correlations between TWS and CGR have not been adjusted for co-varying climate factors such as temperature or VPD.  As a result, these relationships may partly reflect indirect effects mediated by such variables. However, previous studies (e.g., Humphrey et al., 2018) have demonstrated that the influence of TWS on carbon fluxes remains significant and largely independent of temperature, suggesting that the observed coupling is robust despite potential confounding factors.

**lines 543–548: In discussing cases where TWS is weakly correlated with CGR (e.g., tropical Africa in 2016), could consider fire activity, radiation, or phenological anomalies as alternative drivers.**

We will revise the text to acknowledge these alternative drivers and include a brief discussion highlighting their potential influence in regions or periods where TWS–CGR correlations are weak.

**In the conclusion, clearly articulate how your findings can inform terrestrial biosphere model development. For example, suggest that ecosystem models should incorporate regional water constraints with higher fidelity, particularly in tropical forests.**

We appreciate this suggestion and will revise the conclusion to emphasize that terrestrial ecosystem models should prioritize improved representation of water constraints, particularly in tropical forests where water availability plays a critical role in modulating carbon fluxes. As highlighted by Humphrey et al. (2018), current models often underestimate the strength of the coupling between water availability and carbon uptake. Moreover, recent findings by Liu et al. (2023) indicate that this coupling is intensifying over time, further underscoring the need for models to better capture regional water–carbon interactions to improve future projections.

References

Humphrey, V., Zscheischler, J., Ciais, P., Seneviratne, S. I., Friedlingstein, P., Green, J. K., ... & Peters, W. (2018). Sensitivity of atmospheric $CO_2$ growth rate to observed changes in terrestrial water storage. *Nature, 560*(7720), 628–631. https://doi.org/10.1038/s41586-018-0424-4

Keenan, T. F., Hollinger, D. Y., Bohrer, G., Dragoni, D., Munger, J. W., Schmid, H. P., & Richardson, A. D. (2013). Increase in forest water-use efficiency as atmospheric carbon dioxide concentrations rise. *Nature, 499*(7458), 324–327. https://doi.org/10.1038/nature12291

Kleidon, A. (2004). Global datasets of rooting zone depth inferred from inverse methods. *Journal of Climate, 17*(12), 2714–2722. https://doi.org/10.1175/1520-0442(2004)017<2714:GDORZD>2.0.CO;2

Liu, L., Ciais, P., Wu, M., Wang, Y., Walker, A. P., Bastos, A., ... & Poulter, B. (2023). Increasingly negative tropical water–interannual $CO_2$ growth rate coupling. *Nature, 618*(7968), 755–760. https://doi.org/10.1038/s41586-023-06056-x

Saleska, S. R., Wu, J., Guan, K., Araujo, A. C., Huete, A. R., Nobre, A. D., & Restrepo-Coupe, N. (2016). Dry-season greening of Amazon forests. *Nature, 531*, E4–E5. https://doi.org/10.1038/nature16457

---

## Author Comment (AC3)

**"Strong relation between atmospheric CO2 growth rate and terrestrial water storage in tropical forests on interannual timescales" by Petch at al. analyzes CO2 fluxes derived from atmospheric inversions in conjunction with space-based terrestrial water (TWS) to understand what regions drive the accumulation of CO2 in the atmosphere (CGR). Past studies had shown a strong correlation between the atmospheric CGR and global TWS, suggesting that ecosystem water availability is the dominant mediator of the strength of the terrestrial carbon sink and thus TWS. This paper expands on this analysis to show that tropical forests, followed by semi-arid ecosystems, are the drivers of this strong correlation, with tropical America contributing almost 70% of the global TWS-CGR correlation. The manuscript provides some improvement on analysis of globally integrated covariates by elucidating the geographic and ecosystem-type drivers.**

**A major concern with this analysis is lack of discussion of northern extratropics. Previous analysis has shown that these areas are important contributors to CO2 IAV (Guerlet et al., 2013 – who showed that inversions using in situ CO2 only did not see the flux variability that was visible from space-based data; Keppel-Aleks, et al., 2014 – who showed that taking annual averages of CO2 data, which is what is done to calculate CGR in this paper – masked variations that were attributable to northern extratropics). I point out these two papers in particular because the authors use inversions driven by surface CO2 rather than satellite CO2, and because their global correlations are derived from global averaged CO2. Given the four inverse models show a huge spread in the northern extratropical contribution to CO2 IAV, with NISMON showing these areas contribute only 5% to global CO2 IAV and CT2022 showing contributions closer to 35% of global CO2 IAV, more analysis of the impact of this discrepancy is required.**

We thank Reviewer 3 for their thoughtful and insightful comments, particularly regarding the role of the northern extratropics, robustness across inversions, and the interpretation of TWS–CGR correlations. We agree with the need for more discussion. We will explicitly address the NHet, referencing Guerlet et al. (2013) and Keppel-Aleks et al. (2014), and discuss whether some NHet contributions may be muted in CGR derived from surface-based inversions alone.

The analysis in the revised manuscript will now be based on 8 inversion products. (See new figures in response to RC1). The additional products still highlight NISMON as an outlier and support the main results.

**A second concern is that tropical Africa contributes almost as much CO2 IAV as tropical America, but this variability is not as well correlated with variations in TWS. What, then, does this mean for the utility of the high correlation between CGR and TWS? The authors diagnose the regions that drive the apparent global correlation with TWS, but they do not provide much analysis or any conclusions about in what way the emergent global correlation can or should be used in carbon cycle research.**

We will discuss that TWS may not be the dominant constraint in all regions. In tropical Africa, factors such as fire emissions, solar radiation, or phenology may drive CO2 variability independently of water availability. We will expand this discussion and caution against overinterpreting global TWS–CGR correlation as a universal proxy.

**I am also somewhat skeptical how robust are the results from these four inverse models. The models do show substantially different regional fluxes, by at least a factor of two for each of the geographic**

**regions considered. More inverse models are available for analysis, and I am curious about what a larger ensemble would reveal. Section 2.5 stated that correlations were calculated from an ensemble mean flux product. I am curious as to what the correlations with TWS variations would look like from each model separately. Would the conclusions be robust to this change in methodology?**

The contributions to the total land variability were calculated for each inversion product individually, and this has now been done for a larger ensemble size consisting of 8 inversion products. See new figure below. The correlations we have given throughout the study are between regional water storage and the global CGR, which use the GRACE data and the CGR timeseries and do not use the inversions.

[Figure]

**Analysis from the OCO fluxMIP shows significant distinctions between inversions that are configured to adhere tightly to the ocean prior and those where the ocean can move. Are the differences in terrestrial fluxes within this small 4 member ensemble associated with tight ocean priors? And related to a point above, what would the analysis look like based on inversions that use space-based CO2 observations to constrain fluxes? In theory, the space-based data provides better spatial constraints than the sparse surface network.**

We have now expanded our ensemble to include  8 inverse products that fit our time period, all using in situ data only. We agree it would be useful to look at products that use satellite observations as constraints, however, these are only available from 2015 onwards therefore we did not include, as our analysis relies on interannual correlations over a whole 20 yr period. We hope to address this further in collaboration directly with inverse modellers.

**In summary, I recommend more discussion and stronger conclusions about what it means that many regions that contribute a fair amount of CO2 variability do not have a strong correlation with TWS, and other regions that have strong TWS variability do not show much variation in CO2 flux. The paper would have more impact if it provided recommendations about what key carbon cycle inferences are being obscured by using the global relationship between CGR and TWS as a shortcut.**

Thank you for your suggestions. We will add more discussion about points addressed above, and will address what carbon cycle inferences could be missed by using TWS as an indicator. We believe that several of these concerns will be addressed through revisions detailed in responses to RC1 and RC2.

**Guerlet, S., Basu, S., Butz, A., Krol, M., Hahne, P., Houweling, S., et al. (2013). Reduced carbon uptake during the 2010 Northern Hemisphere summer from GOSAT. *Geophys. Res. Lett.*, *40*(10), 2378–2383. https://doi.org/10.1002/grl.50402**

**Keppel-Aleks, G., Wolf, A. S., Mu, M., Doney, S. C., Morton, D. C., Kasibhatla, P. S., et al. (2014). Separating the influence of temperature, drought, and fire on interannual variability in atmospheric CO2. *Global Biogeochem. Cycles*, *28*(11), 1295–1310. https://doi.org/10.1002/2014GB004890**

---

## Author Response (AR1)

Point by point answers to reviewers comments

**RC1**

Comment 1: Throughout the discussion, the authors discuss the findings, but fail to provide some deeper context. In my opinion, the interesting findings of this manuscript should be placed in context better. This would improve the impact of this manuscript, as it allows for follow-up research. I would urge the authors to focus more on the (possible) mechanisms that drive the findings in this manuscript.

For example, the authors do not mention why the tropics account for such a large portion of the covariance. Is this because the IAV in the tropics outweighs IAV in temperate regions (not according to the inversions)? Or is this because droughts in the tropics (i.e. the ENSO cycle) covers the entire tropics (whereas droughts in temperate regions are more driven by synoptic variability and thus happen over smaller scales). Additionally, the authors should mention why TWS is a better explanatory factor than e.g. VPD or temperature. In L.616, the covariance between temperature and water availability is mentioned but not discussed sufficiently.

Response: We have added text describing reasoning to look at TWS rather than other variables L53-57. We have added text about the significant role of the tropics L235-239. We have also added more text about covariance between temperature and VPD L494-495.

Comment 2: Finally, the positive correlation between temperate TWS and carbon growth rate should be mentioned, regardless of the small contribution to the growth rate. Can this be explained physically?

Response: We mention this positive correlation and give possible physical reason such as opposite response to ENSO L219-222

Comment 3: Additionally, the analyses should be done with more than four inverse models. The GCB inverse fluxes are made publicly available

(https://meta.icos-cp.eu/objects/FHbD8OTgCb7Tlvs99IUDApO0 for GCB2023 and more recently https://meta.icos-cp.eu/objects/GpFcABoKcZMVnRUILHRInhdM for GCB2024). Some of these models indeed only cover the OCO2 period (2015 onwards), but for GCB2023 and 2024, 8 systems with sufficient data are provided. Therefore, I expect the analyses to be done with all available models.

Response: We have updated analysis to include all 8 inversions in GBC2023.

Comment 4: L.75: A reader might find it strange that the first sentence of the plain language summary says the CO2 increases every year, but reads here that there is a decrease (which I understand is in the growth rate, so there could still be an increase). I would recommend to maybe rephrase to larger and smaller growth rates with El Nino/La Nina

Response: Plain language summary is no longer included since it is not required for journal format.

Comment 5: L.96: "Most vegetation responds to soil moisture". Yes, but also to VPD (which is not included in TWS). These effects are difficult (if not impossible) to disentangle, but interesting to mention.

Response: We now mention it is difficult to disentangle L58-62.

Comment 6: L.106: I think your definition of NBE is the same as the definition of NEE used here. It could be easier for a general reader to use only one of the two.

Response: In my study I use NBE consistently when referring to my own work. I mention NEE only when referring to other studies who have spoken about 'NEE'. While the two quantities will show similar variability, NBP captures the total flux between land and atmosphere, including disturbances which NEE does not include. Therefore, I believe it is correct to use NBP for the work in my study but still need to use NEE when referencing the other studies.

Comment 7: L.237: Linear detrending removes the fossil fuel trend, but also any trend in biogenic and ocean fluxes. This should be mentioned

Response: This is now mentioned L165-166.

Comment 8: L.238: Why are the CGR and TWS smoothed using different windows? To me it makes intuitive sense to use the same window

Response: Why different smoothing windows are used is now mentioned in text L167-169.

Comment 9: Fig 2: Here, the growth rate can become negative (so the first sentence of the introduction is not true, or is this detrended growth rate?)

Response: This is detrended growth rate. 'de-trended' is stated L194.

Comment 10: Fig 3: The positive correlations in south Brazil and east China are quite interesting, and could be explained biophysically. This links to my first general comment as well.

Response: We mention this positive correlation and give possible physical reason such as opposite response to ENSO L219-222.

Comment 11: L.305: It would be nice to add a horizontal line showing the global correlation to make it more explicit that the tropics can indeed explain all the correlation

Response: Horizontal line has been added.

Comment 12: L.432: It's interesting that NISMON has a larger range, and I recommend the authors to (shortly) discuss any potential reason for this (e.g. prior model, observations used, transport model). The inclusion of other models might help in this.

Response: We discuss NISMON as outlier L335-343.

Comment 13: L.537: I'm not sure I understand the phrasing 'still a key factor'. Is this in contrast to other studies that pointed towards temperature? Otherwise just remove the 'still' Response: removed.

Comment 14: The abstract reports a strong TWS–CGR correlation, but it lacks a statement explaining how this finding advances previous work (e.g., Humphrey et al., 2018). Please clarify further how this study uniquely extends or deepens our understanding.

Response: Abstract revised L17-18.

Comment 15: Lines 28-31: The sentence "tropical forests exhibit the strongest CGR correlations" is important but could briefly explain why—e.g., due to high productivity sensitivity to water stress—so help readers understand the physiological context.

Response: Abstract revised L10-12.

Comment 16: In Section 2.3, clarify whether all four inversion products use harmonized fossil fuel and biomass burning emissions (e.g., GFED versions). Differences in fire emissions datasets could bias regional flux attribution.

Response: Clarified in L155-156.

Comment 17: Table 1 summarizes inversion methods, but the main text should include 2–3 sentences interpreting key differences in transport models, meteorological fields, or prior flux assumptions and how they might affect tropical vs. extratropical estimates.

Response: New paragraph added discussing differences L141-149.

Comment 18: I don't think it's necessary to place Figure 1 in the main text. It is recommended to put it in the supplementary files. Overall, there are too many figures in the full text. It is suggested to combine them.

Response: Figure 1 has been moved to the supplementary.

Comment 19: Lines 293-294: Please explain why some regions with high local correlation contribute minimally to the global signal—e.g., due to small TWS variance or maybe small spatial extent—and explicitly state how these cases are handled.

Response: Explanation added and method to address this given L212-217.

Comment 20: In Figure 4, report the number of grid cells per land cover class and provide standard deviations or interquartile ranges to contextualize variability in contribution estimates.

Response: Regional areas are now given L227-231.

Comment 21: Annotate key ENSO or drought years (e.g., 2005, 2010, 2015–16) directly in Figure 5 to aid interpretation of CGR–TWS relationships and align with the narrative.

Response: ENSO years have been added to Figure 4 (previously figure 5).

Comment 22: lines 390–395: reference a figure or appendix that visualizes cross-regional TWS anomaly compensation (e.g., a correlation matrix or spatial covariance map), supporting the claim of cancellation effects in croplands.

Response: Figures added to supplementary supporting this claim.

Comment 23: The sensitivity analysis in Figure 10 is valuable, but the ecological interpretation of why tropical forests show both high correlation and high sensitivity should be more deeply discussed—e.g., in terms of water-use efficiency or rooting depth.

Response: Deeper discussion of this added L413-421

Comment 24: lines 530–534: Clarify whether TWS–CGR correlations were adjusted for or confounded by co-varying climate factors (e.g., temperature, VPD). If not adjusted, include a cautionary note on potential indirect effects.

Response: Cautionary note added L494-500.

Comment 25: lines 543–548: In discussing cases where TWS is weakly correlated with CGR (e.g., tropical Africa in 2016), could consider fire activity, radiation, or phenological anomalies as alternative drivers.

Response: Added discussion of this L435-440.

Comment 26: In the conclusion, clearly articulate how your findings can inform terrestrial biosphere model development. For example, suggest that ecosystem models should incorporate regional water constraints with higher fidelity, particularly in tropical forests.

Response: Added to conclusion L564-566.

**RC3**

Comment 27: A major concern with this analysis is lack of discussion of northern extratropics. Previous analysis has shown that these areas are important contributors to CO2 IAV (Guerlet et al., 2013 – who showed that inversions using in situ CO2 only did not see the flux variability that was visible from space-based data; Keppel-Aleks, et al., 2014 – who showed that taking annual averages of CO2 data, which is what is done to calculate CGR in this paper – masked variations that were attributable to northern extratropics). I point out these two papers in particular because the authors use inversions driven by surface CO2 rather than satellite CO2, and because their global correlations are derived from global averaged CO2. Given the four inverse models show a huge spread in the northern extratropical contribution to CO2 IAV, with NISMON showing these areas contribute only

5% to global CO2 IAV and CT2022 showing contributions closer to 35% of global CO2 IAV, more analysis of the impact of this discrepancy is required.

Response: We have added more text talking about disagreements in NH et and included the reference Guerlet et. al. (2013) L347-353. We also mention the seasonal sensitivity in extratropics and reference Keppel-Aleks et. al (2014) L467-469.

Comment 28: A second concern is that tropical Africa contributes almost as much CO2 IAV as tropical America, but this variability is not as well correlated with variations in TWS. What, then, does this mean for the utility of the high correlation between CGR and TWS? The authors diagnose the regions that drive the apparent global correlation with TWS, but they do not provide much analysis or any conclusions about in what way the emergent global correlation can or should be used in carbon cycle research.

Response: Text added to discuss this in L444-451.

Comment 29: I am also somewhat skeptical how robust are the results from these four inverse models. The models do show substantially different regional fluxes, by at least a factor of two for each of the geographic regions considered. More inverse models are available for analysis, and I am curious about what a larger ensemble would reveal. Section 2.5 stated that correlations were calculated from an ensemble mean flux product. I am curious as to what the correlations with TWS variations would look like from each model separately. Would the conclusions be robust to this change in methodology?

Response: More inversion products are now included in the analysis

Comment 20: Analysis from the OCO fluxMIP shows significant distinctions between inversions that are configured to adhere tightly to the ocean prior and those where the ocean can move. Are the differences in terrestrial fluxes within this small 4 member ensemble associated with tight ocean priors? And related to a point above, what would the analysis look like based on inversions that use space-based CO2 observations to constrain fluxes? In theory, the space-based data provides better spatial constraints than the sparse surface network.

Response: See response to reviewer 3 in original response posted.

Comment 31: In summary, I recommend more discussion and stronger conclusions about what it means that many regions that contribute a fair amount of CO2 variability do not have a strong correlation with TWS, and other regions that have strong TWS variability do not show much variation in CO2 flux. The paper would have more impact if it provided recommendations about what key carbon cycle inferences are being obscured by using the global relationship between CGR and TWS as a shortcut.

Response: Overall in updated manuscript we have added more in depth discussion in several areas, included a larger range of inversions in analysis, and tried to emphasise more clearly the use of results e.g. L564 -566.

---

## Author Response (AR2)

**Response to reviewer**

Thank you for your re-review. We apologize for not addressing the change in NH results from the additional inverse models more carefully. We agree with the review that two of the additional inverse models indicate higher interannual fluxes from the NH, and some of the regional ensemble numbers from the inverse models were not correctly updated to reflect this.

The main difference between the previous and updated results is a stronger contribution from the NH, due to CTE, IAPCAS and MIROC all having notably higher NH contributions than the previous ensemble mean. The variability across the 8 inverse model results is very large even at continental scales although the ensemble results still support the tropics as the most important region for CO2 flux variability (66%). These results highlight that the ensemble means should not be over-interpreted, as they can be misleading by reflecting cancelling contributions from disagreeing products. We have therefore modified the paper text to emphasise the lack of more spatial consistency however and that better methodology and atmospheric data would be needed to make comparisons with the water inferred regional fluxes. Specifically, modified text discussing the NH inversion results can be found at lines 328–331, 366–369, 495–496, 510–514, and 541–543.

Ensemble numbers that were not previously updated to include all eight products have also been corrected (lines 14, 327, 331, 543).

Other changes include updating Figure 6 to use a shared colour bar and removing some text discussing NISMON as an outlier, as this is now less relevant given the large variability across all products.